# Whole genome deconvolution unveils Alzheimer's resilient epigenetic signature

Eloise Berson [1,2,3] ✉, Anjali Sreenivas[1,2], Thanaphong Phongpreecha [1,2,3], Amalia Perna [1], Fiorella C. Grandi[4,5,6], Lei Xue[2,3,7], Neal G. Ravindra [1,2,3], Neelufar Payrovnaziri[2,3,7], Samson Mataraso [2,3,7], Yeasul Kim[2,3,7], Camilo Espinosa [2,3,7], Alan L. Chang [2,3,7], Martin Becker [2,3,7], Kathleen S. Montine[1], Edward J. Fox[1], Howard Y. Chang [8,9], M. Ryan Corces [4,5,6], Nima Aghaeepour [2,3,7,10] & Thomas J. Montine[1,10]

Assay for Transposase Accessible Chromatin by sequencing (ATAC-seq) accurately depicts the chromatin regulatory state and altered mechanisms guiding gene expression in disease. However, bulk sequencing entangles information from different cell types and obscures cellular heterogeneity. To address this, we developed Cellformer, a deep learning method that deconvolutes bulk ATAC-seq into cell type-specific expression across the whole genome. Cellformer enables cost-effective cell type-specific open chromatin profiling in large cohorts. Applied to 191 bulk samples from 3 brain regions, Cellformer identifies cell type-specific gene regulatory mechanisms involved in resilience to Alzheimer's disease, an uncommon group of cognitively healthy individuals that harbor a high pathological load of Alzheimer's disease. Cell type-resolved chromatin profiling unveils cell type-specific pathways and nominates potential epigenetic mediators underlying resilience that may illuminate therapeutic opportunities to limit the cognitive impact of the disease. Cellformer is freely available to facilitate future investigations using high-throughput bulk ATAC-seq data.

Transcriptional regulation and chromatin accessibility have been shown to play a crucial role in various neurological disorders[1]. Among other epigenetic techniques, the Assay for Transposase Accessible Chromatin by sequencing (ATAC-seq) provides an accurate way to depict the chromatin landscape of the brain and how it is altered by neurodegenerative diseases[2–4]. ATAC-seq is notably relevant to nominate candidates involved in disease, especially non-coding regions that disrupt gene transcription. While bulk ATAC-seq promises to determine open chromatin regions (OCR) and gene regulatory changes in a direct and efficient way, it entangles data from different cell types and obscures cell type-specific information[5–7]. Although single nucleus (sn) ATAC-seq can overcome this barrier[7], it is labor-intensive, expensive, and vulnerable to technical dropout impacting data analysis and interpretation[8].

Deconvoluting bulk sequencing data has been widely investigated, especially for RNA-sequencing[9–14] and recently adapted for bulk ATAC-seq as OCR is better at capturing cell type-specificity than gene expression[12,15–17]. These computational approaches rely on a well-

[1]Department of Pathology, Stanford University, Stanford, CA, USA. [2]Department of Anesthesiology, Perioperative, and Pain Medicine, Stanford University, Stanford, CA, USA. [3]Department of Biomedical Data Science, Stanford University, Stanford, CA, USA. [4]Gladstone Institute of Neurological Disease, San Francisco, CA, USA. [5]Gladstone Institute of Data Science and Biotechnology, San Francisco, CA, USA. [6]Department of Neurology, University of California San Francisco, San Francisco, CA, USA. [7]Department of Pediatrics, Stanford University, Stanford, CA, USA. [8]Center for Personal Dynamic Regulomes, Stanford University School of Medicine, Stanford, CA, USA. [9]Howard Hughes Medical Institute, Stanford University School of Medicine, Stanford, CA, USA. [10]These authors jointly supervised this work: Nima Aghaeepour, Thomas J. Montine. ✉e-mail: eloiseb@stanford.edu

designed signature matrix, using only limited and most distinguishing cell type-specific features to estimate the cellular composition of tissue samples. While this matrix approach can help to resolve spatial single-cell gene expression from bulk RNA-seq[18], the definition of cell type-specific marker remains challenging[14,19]. Recently, a deep learning approach has been proposed to bypass this limitation and directly predict cellular abundance from bulk RNA and microarray expression with high accuracy[13]. Cellular abundance change is a milestone in bulk analysis and has led to new insight into biological mechanisms[13,20]. Yet, it prevents a comprehensive understanding of the chromatin accessibility heterogeneity across cell populations and cell-specific OCR variation in disease, limiting bulk sequencing analysis.

Source separation[21] is a widely studied signal processing paradigm that retrieves the set of individual sources from a mixed signal. One classical application is in audio processing, where one microphone is recording multiple instruments that are playing simultaneously. Source separation consists in retrieving the sound made by each type of instrument individually from one recorded audio signal. In this study, using a similar paradigm, we develop and test a deep learning-based algorithm, Cellformer, that separates the expression of 6 main brain cell types from bulk samples: 4 glial cell types including astrocytes (AST), microglia (MIC), oligodendrocytes (OLD) and oligodendrocyte progenitor cells (OPCs) and 2 major classes of neurons, excitatory (EXC) and inhibitory (INH). Unlike previous studies, Cellformer not only estimates cellular composition but also deconvolutes cell type-specific ATAC-seq OCR along the whole genome.

As we age past 65 years, the majority of the population resides on the Alzheimer's disease (AD) continuum, meaning that approximately four out of five older adults have latent, prodromal, or full expression of AD dementia (ADD)[22,23]. These stages within the AD continuum typically have a progression of functional decline matched with increasing disease burden as measured during life by histopathology, neuroimaging, or biomarkers[24–26]. Standing apart from the AD continuum is a relatively small subset of older individuals who have mismatched normal cognitive function and a high disease burden sufficient to cause dementia; these individuals, called resilient to AD (RAD), are especially important because their existence demonstrates that even advanced AD burden does not necessarily lead to dementia. What combination of inherited factors, life choices, and experiences incurred or avoided combine in this "natural protection" that can be fully effective even in centenarians? Recently, several putative genetic loci involved in RAD have been found using genome-wide association studies (GWAS)[27,28], yet the underlying gene regulatory machinery that mediates gene expression in RAD remains to be elucidated.

In this work, we leverage 191 well-curated tissue samples from sex and age range matched normal control (NC, $n = 5$), RAD ($n = 12$), and ADD ($n = 19$) individuals, without neurological comorbidities, and use Cellformer to predict cell type-specific ATAC-seq data from three brain regions and provide unique insights into the cellular and molecular mechanisms underpinning RAD (Fig. 1, Supplementary Fig. 1).

## Results

### Cellformer: from bulk to cell type-specific OCR

The rich diversity of cell type-specific changes can be obscured in bulk tissue transcriptomic and epigenomic analyses by mixing across heterogeneous cell populations. Hence, we hypothesize that deep learning algorithms, developed to separate mixed source signals[21], could help resolve cell type-specific expression. However, a major pitfall of deep learning is the requirement of large and annotated datasets to train the model without overfitting, yet bulk ATAC-seq datasets with corresponding known cell type-specific expression compositions are very scarce. To bypass this limitation, we leveraged single-nucleus ATAC-seq collected from the brains of NC individuals[7] and an in-silico dataset generation strategy to create synthetic bulk samples with established cell type-specific expression[13] (Fig. 2a). More precisely, Cellformer was trained using synthetic subject-specific synthetic bulk samples. These samples were generated by first sampling and merging a random number of single nuclei per cell type, to create cell type-specific ground truth. Then combining cell type-specific expression produces synthetic bulk samples, Cellformer's input (Methods).

Processing DNA-based sequencing has inherent challenges including handling the extremely large number of sequential features, which can lead to both memory and computational challenges. Deep learning offers the promise of dealing with high dimensional data and showing successful applications in diverse tasks using ATAC-seq data[29,30]. To handle the ATAC-seq high-dimensionality, Cellformer combines attention mechanisms and an effective method, named dual-path. The attention mechanisms create connections between distantly related elements demonstrating high performances in long-sequence modeling with the development of Transformer models[31] in natural language, speech[32], or DNA-sequence processing[33]. The dual-path strategy splits the input sequence into small chunks to extract both local and global dependencies while reducing the computational complexity of attention-based architecture[34]. Applied to ATAC-seq data, Cellformer processes genome-wide sequences extracting both local (intra-chromosome) and global (inter-chromosome) interactions to accurately predict cell type-specific accessibility along the whole genome (Fig. 2b & Methods).

Another well-known issue with ATAC-seq data is low signal intensity[35], which might impact reproducibility and make computational analysis more difficult. To strengthen our model prediction and denoise the ATAC-seq OCR, Cellformer automatically filters the less predictable OCR per cell type, to retain high-confidence OCR for downstream analysis (see Methods).

### Cellformer successfully deconvolutes synthetic bulk ATAC-seq into cell type-specific chromatin accessibility from different tissues

Model validation was carried out using a leave-one subject-out strategy, that is, at each iteration, the training of the model was done using synthetic bulk ATAC-seq brain samples from 12 subjects while

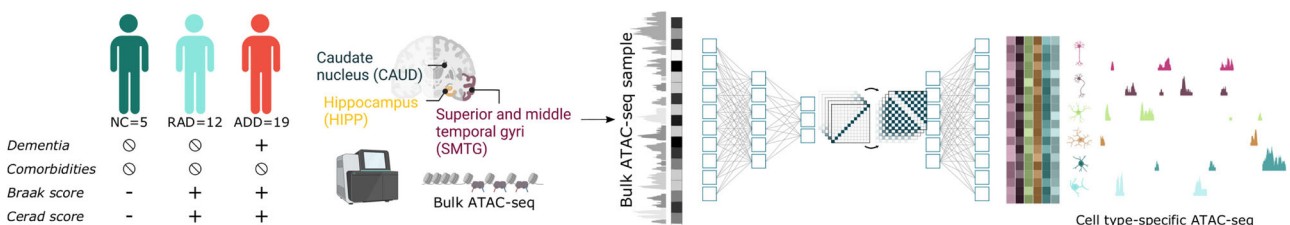

**Fig. 1 | Study overview.** Cellformer was fed data from comorbidity-free bulk samples from individuals with clinicopathologic characterization as normal control (NC), Resilient to Alzheimer's disease (RAD), and Alzheimer's disease dementia (ADD). Three brain regions were used per individual to gain insight into the regional and cellular epigenetic profile of RAD. Cellformer generated cell type-specific expression for 6 main cell types across the whole genome, leading to an unprecedented chromatin profiling of RAD (Created with Biorender.com).

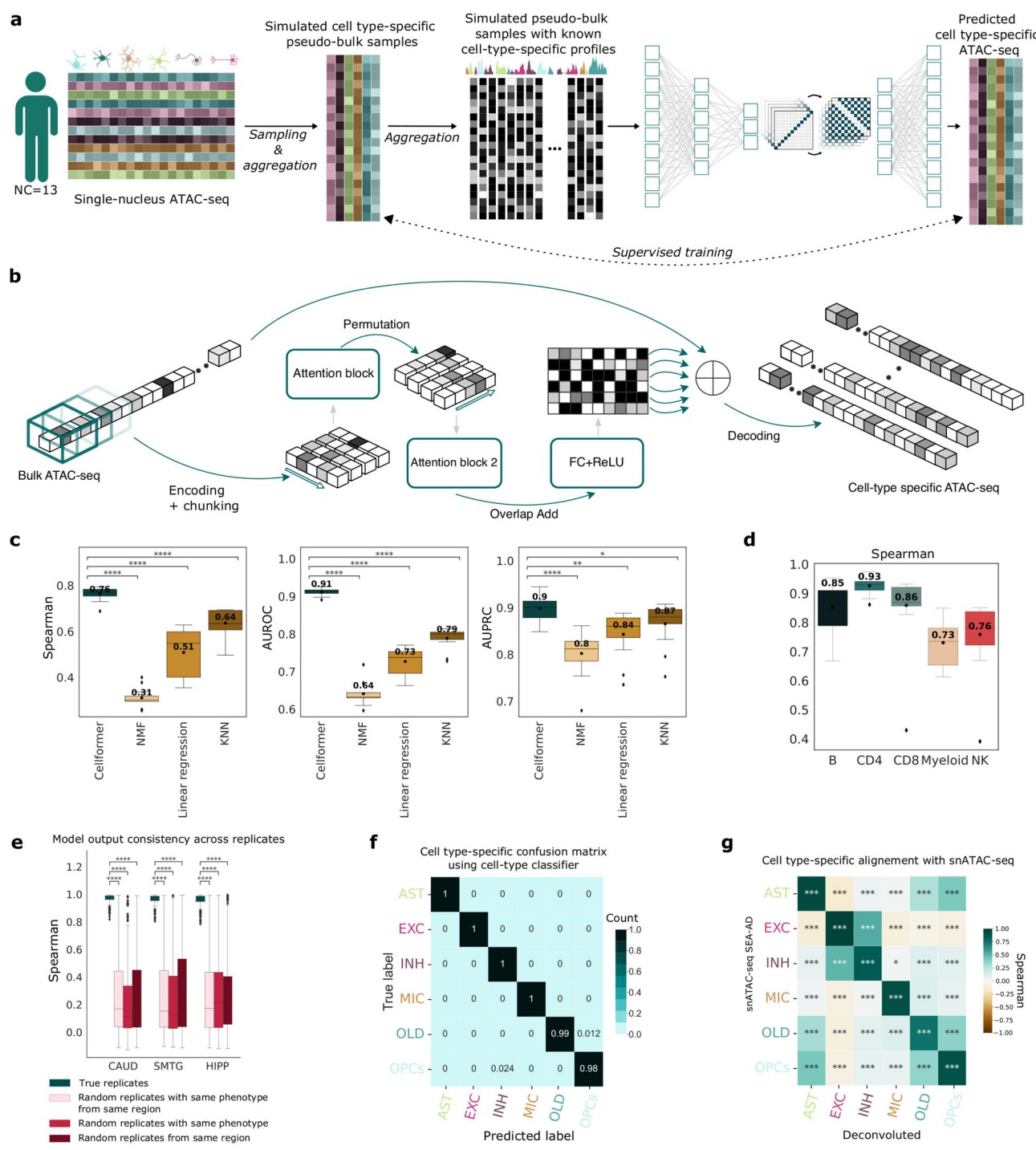

the testing used the 13th subject's sample. The model capacity at accurately predicting OCR value per cell type was assessed using Spearman correlation coefficients. Additionally, the model accuracy at predicting OCR accessibility (yes/no) was measured using AUROC, and AUPRC between binarized ground truth and predicted OCR accessibility (see Methods). Cellformer successfully deconvolutes bulk expression over cross-validation iterations, achieving strong performances with a mean Spearman coefficient of 0.82, AUROC of 0.97, and AUPRC of 0.97 between predicted cell type-specific expression and the synthetic ground truth (Fig. 2c). Stratified by cell type, Cellformer accurately deconvolutes bulk ATAC-seq OCR with Spearman correlation superior to 0.75 (Supplementary Fig. 2a). Cellformer significantly outperforms other machine learning methods chosen as baseline performers for this problem; these include both a supervised and

unsupervised approaches used in partial deconvolution: multi-output linear regression[36,37] and widely used non-negative matrix factorization (NMF)[38] and a nonparametric method, K-nearest neighbors (KNN) (P value < 0.05). Its variance across cross-validation iterations were also lower than existing methods. (Fig. 2c, Supplementary Fig. 2b).

Current state-of-the-art deconvolution methods such as scDeconv[39], DeconPeaker[12], BayesPrism[40], and CIBERSORT[9], rely on a cell type-specific expression matrix, using the most highly distinguished markers per cell type, to predict the cellular composition of bulk tissue. In contrast, Cellformer predicts cell type-specific expression of more than 41954 OCR, which is 2.5-fold more output than established deconvolution methods (Supplementary Fig. 2c). This enables more comprehensive downstream analysis of biological systems at the cell type level and highlights

**Fig. 2 | Cellformer: model training, design, and evaluation. a** A synthetic dataset of simulated bulk samples was generated from previously published single-cell ATAC-seq from 13 normal controls[7]. Cell type-specific pseudo-bulk samples were generated by aggregating snATAC-seq data, revealing the ground truth cell type-specific composition. Simulated cell-specific pseudo-bulk samples were further aggregated to generate pseudo-bulk samples, which are Cellformer's input. This dataset was used to train Cellformer to minimize the reconstruction error between predicted and ground truth cell type-specific ATAC-seq (Created with Biorender.com). **b** Cellformer leverages a dual-path strategy to process both intra and inter-chromosome interaction, enabling full genome deconvolution. *P* values were derived using a two-sided Wilcoxon's test after multi-testing correction. **c** Cellformer was evaluated using the leave-one-subject-out strategy. It outperformed other multi-output regression models, notably linear regression, KNN and an unsupervised approach (NMF) used previously to estimate cellular composition across the (*n* = 6) different cell types. *P* values were derived using a two-sided Wilcoxon's test after multi-testing correction. **d** Cellformer successfully deconvoluted leave-one-out cross-validated PBMC in-silico bulk ATAC-seq data from different datasets (*n* = 18 samples), predicting cell type-specific expression of five main cell types (B cell, T cell-CD4+ (CD4), T cell-CD8+ (CD8), Myeloid and NK cells). *P* values were derived using a two-sided Wilcoxon's test after multi-testing correction. **e** Quality of the Cellformer's predictions was assessed by comparing

technical replicate cell type-specific expression (*n* = 36 samples, see Fig. 1). Cellformer generated outputs that are highly consistent between true technical replicates, exhibiting a correlation coefficient (>0.9) significantly higher than with random replicates. (Two-sided Wilcoxon's test after multi-testing correction) **f** Cellformer output preserves cell type signature across 6 cell types: astrocytes (AST), microglia (MIC), oligodendrocytes (OLD), and oligodendrocyte progenitor cells (OPCs), and 2 major classes of neurons, excitatory (EXC) and inhibitory (INH). An external cell classifier trained on single-cell data from NC samples was used to assess the cell type-specific ATAC-seq quality. The confusion matrix computed between the cell classifier and Cellformer predictions showed almost perfect agreement, highlighting its capacity to preserve the cell type signature. **g** Cellformer validation was performed by comparing RAD cell type-specific expression from SMTG with RAD single-cell ATAC-seq expression from SEA-AD using a two-sided Spearman correlation. Significant high correlations were obtained within the same cell type between the two datasets. Spearman correlation coefficient order between cell types was consistent with biological knowledge: a high correlation was found between neuron types and between OLD and OPCs. All box plots show the median (middle line), interquartile range (bottom and upper edges), and the minimum and maximum values of the distribution (whiskers). *P value < 0.05, **P value < 0.01, ***P value < 0.001, ****P value < 0.0001.

the ability of more extensive deconvolution to gain deeper insight from bulk data.

As a learning-based algorithm, Cellformer relies on snATAC-seq to learn cell type profiles. Using our proposed synthetic pseudo-bulk data generation strategy, we show that Cellformer can be trained effectively with a limited number of snATAC-seq samples, with minimal effect of sample size on its performance (Krustal–Wallis *P* value 0.98)(Supplementary Fig. 2d).

We further test the ability of Cellformer to deconvolute bulk ATAC-seq from different tissues. To ensure its robustness to technical variations such as batch effect, we apply it to in-silico bulk ATAC-seq, which was created from scATAC-seq from 18 peripheral blood mononuclear cells (PBMCs) collected for different investigations[3,41]. Cellformer accurately predicts cell type expression of the five main PBMC cell types with a mean Spearman correlation of 0.85 and minimal cross-sample variation, outperforming other baseline models (Fig. 2d, Supplementary Fig. 2e, f).

In real-life scenarios, the cell type composition of bulk tissue remains unknown. For instance, a rare cell type can be missing, or a new (unidentified) cell type can emerge in bulk tissue. In both scenarios, Cellformer is minimally affected by the presence or absence of one cell type, as there are no significant differences in the model's performance across different cell types (Supplementary Fig. 3). Additionally, we evaluated Cellformer's performances on pseudo-bulk samples made with different percentages of cell type-specific cells. We observed a slight decline in Cellformer's performance when cells make up <10% of the total bulk cells. For biologically rare cells such as OPCs (constituting <3% in white matter) or microglia (constituting <10% in brain), Cellformer achieves an average Spearman correlation of 0.7 when deconvoluting pseudo-bulk data, with OPCs which account for <3% of the overall composition. Similarly, an average correlation of 0.68 is achieved when deconvoluting pseudo-bulk samples containing less than 10% of microglia (Supplementary Fig. 4a). Finally, although we primarily focused on the major brain cell classes in this study, we also assessed the performance of Cellformer in accurately capturing OCRs in specific subclasses such as SST+ and PVAL+ inhibitory neurons (Supplementary Fig. 4b).

### Cellformer resolved bulk ATAC-seq across three brain regions

Following training, Cellformer was then applied to bulk ATAC-seq from NC, RAD, ADD collected from three brain regions: caudate (CAUD), superior and middle temporal gyri (SMTG), and hippocampus (HIPP). Cellformer output consistency, applied to bulk samples from different phenotypes, was done by computing the Spearman correlation

coefficient between technically replicated cell type-specific expressions. A significantly higher correlation (Spearman>0.8, Bonferonni corrected *P* value < 1e-3) is observed between deconvoluted expression from true technical replicates than randomly chosen samples, from the same brain region and same disease group (Fig. 2e, Supplementary Fig. 5). Preservation of the true cell type signature on deconvoluted RAD and ADD samples is evaluated using an external cell classifier, trained on single-cell ATAC-seq from NC (see Methods). A near-perfect concordance is found between the cell-classifier predictions and the true label using Cellformer's outputs (Fig. 2f, Supplementary Fig. 6).

Validation of the RAD and ADD cell type-specific expression was performed by comparing Cellformer cell type-specific expression and cell type-specific expression from snATAC using two publicly available datasets[42,43]. Significantly high correlations (correlation coefficient >0.75) are found between snATAC and deconvoluted cell type expression from two different regions of the cortex using Cellformer's set of predictable OCR (Fig. 2g, Supplementary Fig. 7). A substantial correlation is also noticed between neuronal and glial cells, in agreement with brain cell atlas hierarchy[44]. These inter-cell type correlations were also observed within snATAC-seq and deconvoluted ATAC-seq mean profiles, suggesting that Cellformer can deconvolute cell types with a range of similar OCRs (Supplementary Fig. 7b). These results suggested that the set of predictable OCR derived by Cellformer are highly reproducible across studies and provide a reference signature of the main cell types in the brain that could be a useful resource for further studies.

We next tested whether Cellformer could reveal biological signatures by intersecting AD-specific OCR with genomic regions linked to recently reported AD-risk genes[45]. OCR was derived using univariate analysis comparing ADD with non-ADD samples (adjusted *P* value < 0.05, absolute logFC > 0.5 using two-sided Wilcoxon's test) (Supplementary Fig. 8a). When compared with known AD-risk variants, we found that hippocampal cell type AD-specific OCR associated genes were significantly enriched in AD traits in both neuronal and glial cells, except in oligodendrocytes (*P* value < 0.05 using Benjamini-Hochberg corrected Fisher's test) (Supplementary Fig. 8b). These results highlight the power of cell type-specific resolved ATAC-seq to identify disease epigenetic signatures.

### Cell type-specific chromatin accessibility landscape from NC, RAD, and ADD corroborates previous observations

When applied to bulk samples from NC, RAD, and ADD, Cellformer led to an unprecedented cell type-specific epigenetic dataset offering

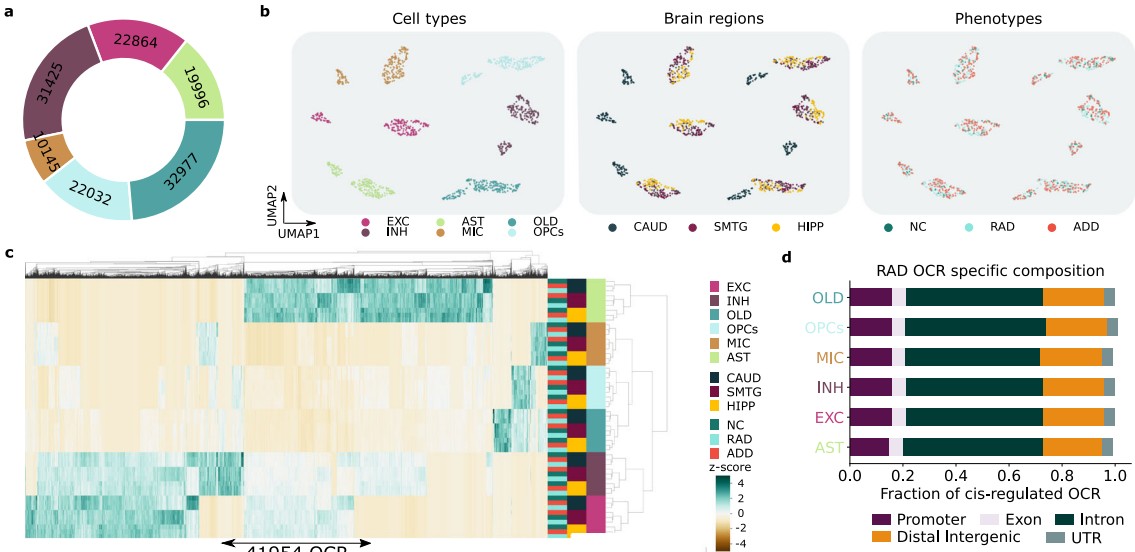

**Fig. 3 | Cellformer deconvolutes bulk expression into cell type-specific expression enabling an unprecedented chromatin profiling of RAD and ADD. a** Our approach enabled the characterization of RAD spanning >10,000 cell type-specific OCR. **b** As expected, cell type-specific expressions were mainly clustered by cell type. UMAP embedding of cell type-specific expression is colored by cell type, brain region, and phenotype. **c** Heatmap showing the mean chromatin accessibility per cell type, brain region, and phenotype. As observed previously[7], phenotype chromatin accessibility variation is dominated by regional variation. **d** OCR composition per cell type. The majority of the cell type-specific OCR were in intronic, promoter, and distal regions.

opportunities to identify unique RAD epigenetic signatures at cell type resolution. It generated at least 10,145 well-predicted OCR per cell type, yielding 41,954 shared cell type-specific OCR, across three brain regions: HIPP, CAUD, and SMTG (Fig. 3a). These OCR exhibited no confounding patterns by sex, age, or batch enabling a more accurate analysis (Supplementary Fig. 8c).

Aligning with the literature, cell type-specific samples clustered primarily by cell type, then brain region, and finally disease group (Fig. 3b, c)[7,42]. Additionally, most of the cell type-specific OCR were found in the intronic (~50%), distal (~25%), or promoter (~15%) regions, similar to our previous work[7] (Fig. 3d). Together these findings demonstrate that cell type-specific data generated by Cellformer showed expected epigenetic patterns in the human brain and aligned well with observations made on single-cell data.

### RAD-specific open chromatin accessibility reveals new epigenetic mediators

We performed univariate analysis on cell type-specific expression from RAD, NC, ADD, independently for each cell type. Most of the differences distinguishing RAD from other groups are found in the HIPP (93%), with few differences observed in the CAUD region (7%) using multi-testing corrected two-sided Wilcoxon's test (*P* value < 0.05, absolute fold-change > 0.5); none were identified in SMTG. RAD-specific OCR are shared between neuronal cells (55%) and microglia (28%) (Fig. 4a).

RAD-dysregulated OCR is primarily cell type-specific, with 30% in excitatory neurons and 22.5% in microglia (Fig. 4b). Interestingly, RAD-specific OCR are found more upregulated than downregulated in HIPP (Fig. 4c). Interpretation of these results is supported by applying Gene Ontology (GO) to the genes related to all identified significantly different OCR in RAD, revealing cell junction, synaptic transmission, and neuronal development signals in neuronal RAD-specific OCR and inflammatory response in microglial RAD-specific OCR (adjusted *P* value < 0.05) (Fig. 4d).

We conducted additional validation of RAD epigenetic signatures by using proteomics data previously collected on the same samples[46]. A weak agreement is observed between proteomic expression and ATAC-seq accessibility with a Pearson correlation of −0.001

(Supplementary Fig. 8d). Only 8% of OCR-related genes show overlap with expressed proteins (Supplementary Fig. 8e). Similar results are observed with RAD-specific OCR-related genes, with 4 out of 40 (10%) genes overlapping with expressed proteins. However, in contrast to the overall sample, our analysis reveals that two (50%) protein-coding genes (*VDAC2* and *PGBP5*) exhibited significant upregulation in RAD at both epigenetic and proteomic levels (Supplementary Fig. 8f).

To complement our analysis and nominate RAD gene regulatory elements, an activity-by-contact (ABC) algorithm was applied to the set of predictable OCR and HiChip from different brain regions[7] to predict regional gene enhancer interactions[47]. ABC model determined 16,320 hippocampal enhancer OCR with 15% showing physical evidence only in this region (Supplementary Fig. 8g). By intersecting the set of predicted enhancers with RAD-specific OCR, we found that <50% of RAD-specific OCR were linked to cis-regulated elements while the rest were found in non-coding regions (Fig. 4e). GO and pathway analysis applied to OCR predicted to be localized in both genic and intergenic RAD-specific enhancers revealed significant enrichment of genes related to chemical synaptic transmission in excitatory neurons, inhibitory neurons, and microglia notably Amyloid Beta Precursor Protein Binding Family A Member 2 (*APBA2*), that modulates AD amyloid precursor protein, and BDNF signaling pathway. These findings corroborate previous analyses performed with microarray protein analysis and animal models[48,49].

## Discussion

Bulk ATAC-seq is an effective and efficient method to measure open chromatin accessibility[2,4,35]. In human brains, ATAC-seq may be favored over RNA-based methods for technical reasons, including greater stability of DNA in post-mortem brains and more comprehensive assessment than in single-nucleus assays[50]. While snATAC-seq offers the attractive opportunity to detect cell type-specific open chromatin accessibility, it is highly impacted by dropout events, making snATAC-seq analysis more challenging and vulnerable to missing low-expressed genes[51]. To remedy this, we developed Cellformer, a new approach to deconvolute bulk ATAC-seq data and thereby computationally enhance resolution to the cell-type level. Using Cellformer, we illustrated the power of deep learning to enhance biological data

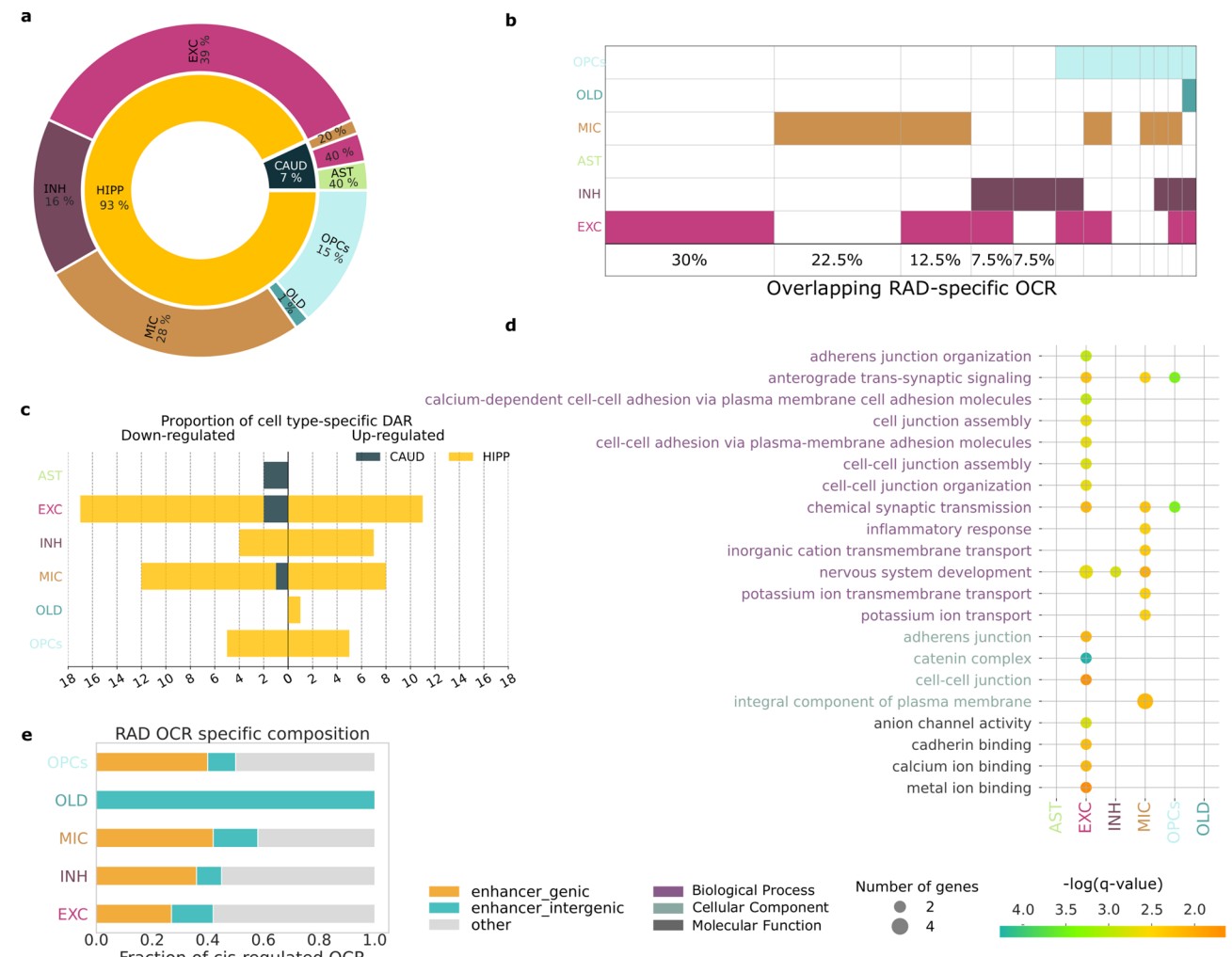

**Fig. 4 | Epigenetic signature of RAD.** Differentially expressed open chromatin region (OCR) across three brain regions and predicted for six cell types. **a** Cell type-specific OCR between RAD and ADD/NC were mainly found in HIPP (93%) and distributed between microglia (28%), and neuron cells (55%) (multi-testing adjusted two-sided Wilcoxon's test *P* values < 0.05, logFC > 0.5). **b** OCR differentially regulated in RAD in the HIPP was mainly neuron-specific (excitatory specific or shared between inhibitory and excitatory neurons) or microglia-specific. **c** Number of differentially upregulated and downregulated OCR in RAD per cell type. **d** GO enrichment applied to RAD-specific OCR (FDR 5%). **e** Hippocampal RAD-specific OCR intersected with predicted enhancers using the ABC model[47]. Only 50% of RAD-specific OCR were identified as enhancers in HIPP.

analysis and advance our understanding of disease mechanisms at the epigenetic level.

The cellular composition of tissue is a crucial component of sequencing analysis[40,52]. In contrast to previous methods to deconvolute bulk sequencing data[9,12,53], Cellformer does not aim to predict cellular abundance in bulk tissue but instead "fully" deconvolutes bulk ATAC-seq, generating OCR data at the cell type level across the whole genome. Besides, Cellformer does not rely on cell type signature matrix definition[9,18], a key ingredient of previous bulk deconvolution that strongly correlates with prediction accuracy[14,19] and deters application to samples where no single-cell data are available, such as the RAD and ADD samples investigated here. Defining an accurate signature remains an open computational problem that has been poorly investigated for ATAC-seq data[12]. Cellformer bypasses this issue by leveraging the power of deep learning to automatically extract and predict cell type-specific OCR[13].

While ATAC-seq can be performed on experimentally isolated single cells or single nucleus, Cellformer can resolve bulk expression at the cell type level, not at the single-cell level. However, creating cell type-specific mixtures is a popular strategy to overcome the low count and sparsity of snATAC-seq and improves the statistical power of single-cell analysis[54]. Notably, cell type-specific data are used to strengthen the signal and improve statistical significance for high-confidence, differential analysis[55], TF footprint, or disease gene regulatory signatures identification[56–58]. Furthermore, similar to single-cell sequencing, Cellformer is limited to the most predictable and highly expressed cell type-specific OCR.

Cellformer belongs to the reference-based method category: supervision of the model requires snATAC-seq to learn cell type-specific expression. Therefore, Cellformer predictions are limited to the cell type-specific open chromatin patterns detected in the single-nucleus samples and strongly depend on the quality of snATAC-seq samples.

In this study, we focused on the 6 major brain cell classes. Although we demonstrate Cellformer's ability to deconvolute at a lower resolution (Supplementary Fig. 4b), we notice that increasing the number of output cell types results in a significant rise in computational complexity. This limitation restricts the number of deconvoluted cell types that can be effectively handled. To overcome this challenge, we plan to implement and validate strategies such as hierarchical training or cell-type prioritization functions, which will expand the capabilities of Cellformer. The model generalization and

robustness will be also improved in the future by leveraging the ever-growing amount of available single-cell/nucleus ATAC-seq data or by adapting transfer learning approaches to improve prediction performances on bulk samples from new tissues. Another limitation is the time of training which varies between two hours and a few days depending on the model configuration, the computing power used for the training, and the number of samples. Yet, it remains faster and less expensive than a single-cell sequencing protocol. Additional hyper-parameter optimization and gradient acceleration strategy will be implemented in the future to improve training efficiency[59].

Applied to large cohort bulk ATAC-seq data from multiple human brain regions, Cellformer provided new insights into RAD, an unusual group of individuals who do not succumb to the high burden of AD and who likely hold important clues to treat this highly prevalent disease. Cellformer predicted that OCR differences between RAD and the other two groups that are on the AD continuum were very strongly localized to HIPP, which subserves declarative memory formation and is the primary target of AD. At the cellular level, most predicted RAD-specific OCR were characterized by changes in both inhibitory and excitatory neurons followed by microglia[60]. From the perspective of cellular processes, GO analysis of RAD-specific OCR highlighted neuronal development, inflammatory response, and synaptic transmission processes. These pathways were highlighted in previous studies using proteomics and mouse models of AD[48,61–63]. Overall, these highly plausible predictions suggest that individuals with RAD are distinguished from the AD continuum by epigenetic upregulation in support of hippocampal neuronal processes and synapses. This regulation change might confer RAD the ability to preserve the number of neuronal projections and synapses that have been observed through histopathological studies[64].

Cellformer offers new perspectives to gain insight into bulk sequencing and identify cell-specific gene regulatory changes in disease progression in a cost-effective way. Complementing cellular abundance prediction, Cellformer provides additional information to advance bulk ATAC-seq analysis. We expect that Cellformer may help to unveil cell-specific transcriptional regulation and advance our understanding of disease epigenetic mechanisms in other biological settings.

## Methods
### Data overview
This study drew on snATAC-seq and bulk ATAC-seq data previously collected[7]. Primary brain samples were obtained from Stanford University, the University of Washington, or Banner Health from postmortem tissue following informed consent and Institutional Review Boards approval[7] (Supplementary Fig. 1). In addition, five recently collected control snATAC-seq samples from SMTG (collected using the same ATAC-seq protocol[7]) were used to train the model. Data on sex was collected but not gender. Validation was performed using Seattle Alzheimer's Disease Brain Cell Atlas (SEA-AD)[43] single-cell (sc-) ATAC-seq data generated using a 10xMultiome preparation, snATAC-seq from human prefrontal cortex[42] and genetic variants from the most recent GWAS study for late-onset Alzheimer's[45].

Individuals' brain samples from both ATAC-seq datasets were carefully filtered according to clinical diagnosis of cognitive status proximate to death and assessment of AD neuropathologic change and other neuropathologic comorbidities (see Supplementary Table 1) using current consensus guidelines[24,65–69]. Resilient cases were defined as individuals without dementia at their most recent clinical research evaluation within 2 years of death, and neuropathologic findings of B score >2 and C score >1 but without vascular brain injury or Lewy body disease, and LATE neuropathologic change stage of 0 or 1. To ensure a valid comparison between cell-type-specific ATAC-seq and single-nucleus ATAC-seq from SEA-AD cohort, we also removed samples from patients with known Lewy Body disease (brainstem, limbic,

neocortical, olfactory) and LATE neuropathologic change stage above 1 in the SEA-AD dataset.

### Bulk ATAC-seq and scATAC-seq processing
We leveraged annotated single-cell ATAC-seq from 12 NC subjects to identify cell type-specific OCR (or peak)[7]. "Peak calling" was performed on regional and cell type-specific replicates to improve statistical significance using ArchR workflow[56] and its MACS2 implementation[70]. Chromatin accessibility varies largely per brain region and cell type[1]. Therefore, to ensure capturing OCR that is significantly expressed in the 6 main cell types of interest (astrocyte, microglia, oligodendrocyte, OPCs, excitatory, and inhibitory neurons), we removed single cells that do not belong to one of these six categories as well as single cells identified as doublets. Only the significant OCR marker[56], unique to an individual or a small number of cell type groups was conserved for downstream analysis (FDR < 0.001 and FC > 1). In total, we defined a set of 41,954 OCR. The count normalized matrix, combining OCR from all the bulk samples, was then derived using featureCount[71].

Previously preprocessed and annotated PBMC scATAC-seq was downloaded from (https://github.com/GreenleafLab/ArchR_2020)[56]. Peak calling, peak filtering, and in-silico bulk generation were performed using the same workflow as described above.

FeatureCount (version 2.0.3) also was applied to 10xMultiple single-nucleus ATAC-seq fragment files from SEA-AD cohort to extract the same set of OCR. For an accurate comparison, we only considered predictable cell type-specific OCR for comparison, used in this study. Only cells passing the quality control and annotated using the paired snRNA were used for downstream analysis[43].

Annotated raw OCR count matrix provided by Morabito et al[42]. was used and intersected with our set of predictable OCR to compare the prefrontal cortex from ADD and NC single nucleus with deconvoluted cell type expression using "intersect" function from bedtools[72]. A set of 20060 overlapping OCR was then used to compute the Spearman correlation matrices between cell type expression (Supplementary Fig. 8).

### Transformer-based cell-specific ATAC-seq separator
The "Cocktail party" or source separation problem is a widely studied question consisting of extracting individual source signals from a mixed one. Inspired by this paradigm, we leveraged a state-of-the-art source separation method to deconvolute bulk ATAC-seq and extract individual cell type-specific expression along the whole genome. Comprehensive processing of the gene regulatory elements linkage, spanning the whole genome, was achieved using a long-sequence friendly neural network developed for speech separation[34]. Based on the inner/outer transformer-based architecture, our neural network can extract both within and cross-chromosome epigenetic dependencies along the whole genome using a reduced number of trainable parameters. More precisely, this "dual-path recurrent neural network" strategy[73,74] decomposes long sequences into smaller chunks of size 250 and, extracts high-level representations within chunks, which are then concatenated and permuted for inter-chunk interaction processing. Dual path-based models have shown to be effective at modeling very long sequences, leading to superior predictive performances in various audio processing tasks. We adapted the published model to predict the ATAC-seq profile of 6 major brain cells including astrocytes, microglia, oligodendrocytes, OPCS, and two subclasses of neurons, excitatory and inhibitory neurons. An advantage of this architecture is that it can extract both within and cross-chromosome cell type-specific epigenetic dependencies all along the genome.

To improve our prediction confidence and enhance the robustness of our approach, we stacked on top of our trained network a filtering module removing predicted OCR with a relatively high training error. For each cell type $i$, we computed the normalized mean absolute error across all the $N$ samples in the training set as:

$\mathrm{NMEA}_p = \frac{\mathrm{MEA}(\hat{\mathbf{X}}_i, \mathbf{X}_i)}{\frac{1}{N}\sum_{p=1}^{N}|\mathbf{X}_i|}$, with $\mathbf{X}_i$ and $\hat{\mathbf{X}}_i$ the ground truth and the predicted, OCR values, respectively for the cell type $i$. We preserved cell type-specific OCR for downstream analysis with a mean error lower than a threshold defined as: $\mathrm{NMEA}_p < M_{\mathrm{NMEA}} + \tau^*\sigma_{\mathrm{NMEA}}$, with $M_{\mathrm{NMEA}}$ and $\sigma_{\mathrm{NMEA}}$ the mean and standard deviation of NMEA across all the OCR and $\tau$ the threshold ranging from −0.5 to 1. $\tau$ was optimized using samples from the training to maximize the correlation between the cell type-specific ground truth and predicted ATAC-seq expression. Once the filtering module was trained, it was applied to the model outputs to preserve only the most predictable cell type-specific OCR.

The pythonic implementation from Asteroid library[75] was used to build and adapt a network. Intra-inter dual-path block was repeated once and comprised 1 multi-head attention layer and 256-dimensional FC layers, leading to a 435 K trainable parameter-neural network. The model was trained using AdamW Optimizer from PyTorch (v1.10.0)[76] to minimize the mean-square error loss on batches of size 32. An initial learning rate of 1e-3 was dynamically optimized during the training using the strategy proposed by[73]. Best iteration and optimal weights were selected using an early-stop algorithm. The training stability was ensured by using gradient clipping to limit the MSE error to 5.

## Synthetic dataset generation

Model training was achieved by creating a synthetic dataset of paired bulk and corresponding cell type-specific samples (ground truth), leveraging available single cells samples from NC[12,13] Each pair in the synthetic dataset was created by first sampling and aggregating a random number of the same type of single nuclei from a sample's snATAC-seq results in order to create synthetic cell type-specific pseudo-bulk samples that preserve regional and individual diversity in our synthetic dataset. Then, corresponding synthetic bulk ATAC-seq data were created by aggregating the generated cell type-specific pseudo-bulk samples from the six cell types. 3000 pairs of synthetic bulk and cell type-specific bulk were generated from each subject, composed of a random number of cells ranging from 100 to 800. Then, both synthetic cell type-specific and bulk samples were normalized by the total number of cells and the maximum OCR value. Harmony batch normalization was applied on snATAC-seq to ensure robust peak calling[7,56]. No batch normalization was further applied on the input of the model, since additional analysis suggests that Cellformer removes batch effects while preserving biological variations, i.e., kBet = 0.79 ± 0.14, NMI = 0, ARI = −0.02 ± 0.001, cLISI = 1, iLISI = 0.58 ± 0.02 across the brain regions[77].

## Model validation

**Model testing through leave-one-subject-out cross-validation.** Model generalizability was assessed using the leave-one-subject-out strategy. More precisely, at each iteration, the dataset was partitioned into a training and testing set, such that samples from all brain regions from one individual were left out while the rest of the samples were gathered to create the training set. From the training set, 20% of the samples were used to tune the model hyperparameters at each iteration while the remaining samples helped to optimize the weights of the model. Once trained, we fed the model with synthetic bulk samples created by aggregating single cells per snATAC-seq donor from the test set (never seen by the model) and validated its performances at predicting an accurate and consistent cell-specific signal using the Spearman correlation. We also assessed the ability of the model to predict non-zero OCR by computing the AUROC and AUPRC after binarization of the ATAC-seq expression[29,30]. Mean errors with quartile error bars across iterations were reported and compared with other models. The model with the highest performance on the whole dataset and among the top three models with the lowest test error was used for downstream analysis.

**Model's output consistency and plausibility.** Model output consistency was validated by computing the Spearman correlation between technical replicates and predicted cell type-specific signals (Fig. 2e). To assess the significance of the observed mean correlation between technical, random replicate permutation tests were performed. More precisely, for each bulk sample, Spearman correlation was computed between the model's output of this sample and a random replicate, arbitrarily selected from the same brain region, from the phenotype group, or both the same brain region and phenotype group (Supplementary Fig. 3). $P$ value was derived by comparing the mean correlation between true replicates and random replicates using Bonferroni corrected two-sided Wilcoxon test.

**Cell signature preservation.** We ensured the cell type signatures model's preservation in ADD and RAD samples using an external cell classifier (Supplementary Fig. 6). To better capture the cell type-specific signature and be more robust to dropout, an XGBoost classifier was trained to classify single-nucleus ATAC-seq from Control cases into cell class. The model was trained to minimize a softmax loss between the predicted label from synthetic cell type-specific ATAC and the corresponding ground truth (see Synthetic Dataset Generation). Stratified K-fold nested cross-validation was exploited to validate the model and performances of the model were quantified using AUROC and AUPRC between the ground truth label and the predicted probability and optimized the model hyperparameters. The XGBoost package in Python was exploited to implement the cell classifier, parameterized with a learning rate equal to 0.1, a maximum depth of tree set to 10 and 100 estimators. The model achieved a mean AUROC = 1.000, Precision = 0.994, and Recall = 0.993 over cross-validation iterations when tested on the held-out single-nucleus ATAC-seq datasets.

Once trained and validated to accurately predict the cell type, the model was applied to deconvoluted cell type-specific expression from AD and RAD. The classifier achieved an AUROC of 1.000, a Precision of 0.993, and a Recall of 0.994 when tested on the deconvoluted expression. Confusion matrices showed almost a perfect classification across cell types, brain regions, and conditions.

## Model comparison

We compared our model with supervised (Linear regression), non-parametric (KNN) machine learning, and unsupervised (Non-Negative Matrix Factorization) models. These algorithms were implemented using the default parameterized functions from Scikit-learn[78]. Using 100 synthetic bulk samples, the multi-out models (Linear regression and KNN) were trained to predict cell type-specific OCR by minimizing the MSE loss. The same leave-one-subject-out strategy as for Cellformer was used to assess models' generalization and avoid overfitting. NMF model was trained using a synthetic bulk matrix created by aggregating all the single nuclei per replicate. Then, the predicted OCR-specific expression was computed through row-wise multiplications between the feature matrix and the coefficient vectors.

## OCR annotation

Chipseeker[79] was used to identify OCR-gene association and genomic OCR annotation using default parameters following ATAC-seq data processing guidelines and Harvard bioinformatics recommendations[35,80]. One of the main issues in epigenetic analysis is the lack of consensus between annotating tools[81]. We, therefore, compared Chipseeker to the annotations given by ArchR, developed for ATAC-seq data analysis. Overall, Chipseeker and ArchR agreed on 60% of peaks, including complete (100%) agreement between peak-to-gene annotations of RAD-specific promoters, from which our biological insights were drawn. In particular, similar gene ontology enrichment is observed using both tools.

Candidate enhancer OCRs per brain region were computed using Active-by-Contact (ABC) model[47]. Candidate enhancer regions were derived for each brain region independently using 10 ATAC-seq replicates. Region-specific ABC scores were computed by combining the OCR activity and the genomic spatial information extracted from the HiChip-seq data provided by Corces et al[7]., using the suggested parameters[80].

## Cell type-specific ATAC-seq analysis

Differentially expressed OCR were identified using FDR corrected two-sided Wilcoxon test using FDR 5% and absolute log fold-change superior to 0.5 from Scanpy library and MultiPy[82–84]. Gene ontology and pathway analysis were performed on OCR-related genes with the GO and BioPlanet databases from 2021 and 2019, respectively, using GSEAPY[85] (version 1.0.3).

## Reporting summary

Further information on research design is available in the Nature Portfolio Reporting Summary linked to this article.

## Data availability

All data supporting the findings described in this manuscript are publicly available. Bulk ATAC-seq and single-cell ATAC-seq from control individuals were previously collected and annotated[26] accessible through GEO accession (GSE147672). Additional single-nucleus ATAC-seq data and raw and processed bulk ATAC-seq from ADD and RAD are available through GEO accession (GSE226529) and Dryad (https://doi.org/10.5061/dryad.2fqz612t0). Validation of the model was performed using processed snATAC-seq from[40,41] available at (http://portal.brain-map.org/explore/seattle-alzheimers-disease and https://www.synapse.org/#!Synapse:syn22079621/wiki/603535. Processed PBMC ATAC-seq data are accessible at https://github.com/GreenleafLab/ArchR_2020).

## Code availability

For future research, all custom code used in this work code, processed data, and additional metadata have been made publicly available at (https://github.com/elo-nsrb/Cellformer) and https://doi.org/10.5281/zenodo.8175353. The following packages were used: Python 3 (version 3.9.7) with PyTorch (version 1.10.0); Scikit-learn (version 1.0.1), asteroid (0.5.2), and GSEAPY (version 1.0.3); R (version 4.2.2) with ArchR (R version 4.2.2), Chipseeker (version 1.36.0); FeatureCount (version 2.0.3).

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

## Acknowledgements

This work was supported by RF1 AG053959 (T.J.M.) U19 AG065156 (T.J.M.) R35GM138353 (N.A.), RF1 AG077443 (T.J.M., N.A.) U01 AG072573 (T.J.M.), NIH RM1-HG00735 (H.Y.C.). H.Y.C. is an Investigator at the Howard Hughes Medical Institute. Schematics were created with BioRender.com.

## Author contributions

E.B., E.J.F., M.R.C., N.A., and T.J.M. conceived and designed the experiments. T.J.M., N.A., M.R.C., E.J.F., K.S.M., and H.Y.C. contributed to the funding acquisition. Samples were acquired by T.J.M., M.R.C., and H.Y.C., E.B. and A.S. performed the experiments with the help of T.P., T.J.M., N.A., M.R.C., A.P., N.G.R., L.X., N.P., S.M., Y.K., C.E. and A.L.C. E.B., A.S., A.P., F.G., L.X., N.G.R., N.P., S.M., Y.K., C.E., A.L.C., and M.B. contributed to data processing and analysis, visualization and analysis tools. E.B., N.A., K.S.M., and T.J.M. wrote the manuscript with the help of all the authors. All cohorts contributed to the manuscript review and editing.

## Competing interests

H.Y.C. is a co-founder of Accent Therapeutics, Boundless Bio, Cartography Biosciences, and Orbital Therapeutics, and is an advisor of 10x Genomics, Arsenal Biosciences, Chroma Medicine, and Spring Discovery. The remaining authors declare no competing interests.
