## [Peer Review File · Nature Communications]

REVIEWER COMMENTS

Reviewer #1 (Remarks to the Author):

The authors present a new deep learning method, Cellformer, that conducts deconvolution of bulk ATAC-seq into cell type-specific expression across the whole genome. Cellformer integrates attention mechanisms that establish links between elements that are far apart and a powerful method called dual-path, which involves breaking down the input sequence into smaller segments to capture both local and global dependencies while also reducing the computational demands of attention-based architectures. Applying Cellformer on 191 bulk samples from three regions of the brain, specific gene regulatory mechanisms associated with certain cell types and potential mediators implicated in resilience to Alzheimer's disease (AD) are identified.

This manuscript is nicely written. And the proposed method, Cellformer, has achieved OK performance in Spearman correlation coefficient, AUROC, and AUPRC on the synthetic bulk ATAC-seq. However, to improve the confidence in Cellformer's performance, the authors should conduct more experiments. Specific comments are as follows:

- (1) In figure 2c, the evidence provided to support the improvement in Cellformer over existing algorithms is given over all cell-types present in the synthetic bulk ATAC-seq brain samples. While this is a good comparison in evaluating a method overall, it does not provide any information about how Cellformer performs on different cell types.
- (2) Since lack of ATAC-Seq data of cell type proportion-known mixture samples, it is reasonable and essential to conduct evaluations on comprehensive synthetic datasets. However, the authors only generate via the setting '5000 pairs of synthetic bulk and cell-type specific bulk were generated from each subject, composed of a random number of cells ranging from 100 to 800'. What happens if certain cell types are missing from the bulk or single-cell data?
- (3) How does Cellformer perform on cells that have only 5% and 2% out of all the samples? One of the main unresolved issues is a good performance on rare cell types.
- (4) Since Cellformer uses reconstruction loss to aid in learning, how does the sample size of the data affect the performance?
- (5) The considered benchmark methods are limited and are rather simple. Moreover, the authors only compare the performance of Cellformer to other benchmarks in Figure 2, but failed to incorporate any comparison in Figure 3 and Figure 4. From my knowledge, the idea of cell-type specific 'deconvolution' is not new in this area; maybe the author should consider conducting comprehensive evaluations with existing works like scDeconv, DeconPeaker, CoRE-ATAC, and references therein.

Reviewer #2 (Remarks to the Author):

Comments for Author

General comments:

In this manuscript, the authors have developed a novel approach for deconvoluting bulk ATAC-seq data to infer cell type-specific ATAC-seq expression as well as sample composition. The authors use a reference-based deep learning method, which they show outperforms other machine learning methods for this task. On the biological side, they compare the cell type-specific expression between resilient to AD (RAD), normal control, and AD dementia (ADD) individuals, and find that open chromatin regions (OCRs) differences between RAD and the other groups are strongly localized to the hippocampus region, and identify key pathways and cell types whose genes may have altered accessibility.

Overall, the manuscript presents a promising approach for deconvoluting bulk ATAC-seq data to infer

cell type-specific ATAC-seq expressions. However, the manuscript still leaves some key questions unanswered, and some additional validation is required to fully support the authors' conclusions.

Major comments:

1. The schematic of the model in Figure 2 could use a little more detail. In particular, it shows the reconstruction of cell type-specific profiles, but does not show where/how the cell type composition measures are derived. Some of these details are in the methods, but it would be good to show them up front, particularly for non-experts.

2. It would be helpful if the authors could provide more details about the batch effects that were adjusted for in their processing pipeline. This could include information about the experimental design, such as the sequencing batches, sample collection dates, and any other relevant factors that may introduce batch effects. The methods used to adjust for batch effects should also be described in more detail. For example, did the pre-processing or model incorporate any batch normalization techniques, such as ComBat or SVA, to adjust for batch effects? If so, they should describe how these methods were applied and any parameters that were used in the process. In addition, the authors should include quantification of batch mixing (e.g. LISI scores or other measures on the plots shown in Fig. S5) to show that batch effects were essentially removed by the model.

3. While the Cellformer approach appears to be effective at deconvoluting bulk ATAC-seq data to infer cell type-specific ATAC-seq expressions, it may not be able to fully capture the cell type subpopulations that are available through single-cell sequencing methods. Is Cellformer limited to the resolution of major cell classes in the brain, or can it also recapitulate OCRs in distinct subtypes within a class e.g. SST+ versus PVAlb+ neurons in cortex and hippocampus? Although it is not necessary to test every subtype, it is possible that the Cellformer approach could be rerun with finer resolution clusters in the training set to identify composition/OCR profiles of subpopulations of cells within bulk sequencing data?

4. The authors used ChIPseeker to annotate peaks to genes may not be the most precise method, as it relies on assigning regions to genes based on proximity. This can lead to false positives or false negatives in the annotation of peaks to genes, as well as potential ambiguities in the assignment of peaks to specific genes. How robust are the biological results regarding RAD robust to different peak-to-gene annotations?

5. It is important for the authors to explore the biological implications of the observed OCR differences between RAD and the other groups. While the authors have performed functional annotation and pathway analysis to identify enriched biological pathways or functions that are associated with the differentially expression open chromatin regions. The authors could perform further validation studies to confirm the significance of the observed OCR differences, for example through staining or ISH of a few predicted cell type-specific genes in the hippocampus tissue. Overall, validating the key biological implications of the observed OCR differences would provide a more complete picture of the mechanisms underlying resilience to Alzheimer's disease, and could have important implications for the development of novel therapeutic approaches.

6. In Fig. 2G, the cross-comparison between model predicted OCRs and SEA-AD OCRs shows some off-diagonal correlations. These are likely biological, given cell type similarity, but it would be good to have "baseline" heatmaps of the same type showing cross-cell type correlations within the SEA-AD data alone and the Cellformer predictions alone. In the likely event that this looks very similar to the existing Fig. 2G, it would strengthen the argument that Cellformer can deconvolute cell types with a range of similar OCRs.

Minor comments

1. Figure 2 would be improved by increasing the asterisk size in panels C, E and G.

2. It would be highly informative if the authors could provide the age, sex and ethnicity information of the RAD, NC and ADD donors the authors used in this study. Although Figure S5C shows there is no strong sex effect at the top cell type level, it would be good to know the balance between sexes, as well as the age distributions between the three groups.

3. In Figure S5C, suggestion to replace "Expired Age" with "Age at death"

4. In the methods (line 461): "For each pair in the synthetic dataset, a random number of same type single nucleus within was merged using single nucleus data from one snATAC-seq, 462 preserving regional and individual diversity in our synthetic dataset", it seems there are some words missing. "single nucleus samples"? "one snATAC-seq donor"?

Response to Reviewer's comments.

All the authors wish to thank the reviewers for their helpful suggestions and for the opportunity to improve our manuscript. As detailed below, we have carefully modified the manuscript in response to every point. All changes are in blue in the manuscript unless otherwise noted. We have performed additional experiments to answer the reviewer's concerns. The only change to the main Figures is that additional details have been added to the schematic of the model (Figure 2A) in response to R2, Point 1 and asterisk size is increased (R2, minor point 1).

Please note that in this table, new figures are bolded and revised figures are underlined.

Original Figure Number	Revised Figure Number
-	Supp Figure 1 (added two panels showing sex/age data distribution in the study)
Supp Figure 1	Supp Figure 2 (added new panels A, B, C and D: Cellformer performance across cell-types, comparison with other previous methods and sample size effect on model's performance).
-	Supp Figure 3-4 (added new evaluation results)
Supp Figure 2	Supp Figure 5
Supp Figure 3	Supp Figure 6
Supp Figure 4	Supp Figure 7 (added panel B: cross-cell type baseline heatmaps)
Supp Figure 5	Supp Figure 8 (added new panels D,E and F: validation on external proteomics data)

Point-by-Point Revisions to Reviewers' Critiques

Reviewer #1 (Remarks to the Author):

The authors present a new deep learning method, Cellformer, that conducts deconvolution of bulk ATAC-seq into cell type-specific expression across the whole genome. Cellformer integrates attention mechanisms that establish links between elements that are far apart and a powerful method called dual-path, which involves breaking down the input sequence into smaller segments to capture both local and global dependencies while also reducing the computational demands of attention-based architectures. Applying Cellformer on 191 bulk samples from three regions of the brain, specific gene regulatory mechanisms associated with certain cell types and potential mediators implicated in resilience to Alzheimer's disease (AD) are identified.

This manuscript is nicely written. And the proposed method, Cellformer, has achieved OK performance in Spearman correlation coefficient, AUROC, and AUPRC on the synthetic bulk ATAC-seq. However, to improve the confidence in Cellformer’s performance, the authors should conduct more experiments. Specific comments are as follows:

(1) In figure 2c, the evidence provided to support the improvement in Cellformer over existing algorithms is given over all cell-types present in the synthetic bulk ATAC-seq brain samples. While this is a good comparison in evaluating a method overall, it does not provide any information about how Cellformer performs on different cell types.

Response: We added information about how Cellformer performs on different cell types as new panels A and B of (now) Supplementary Figure 2, and introduced it in the Results section as: “Stratified by cell type, Cellformer accurately deconvolutes bulk ATAC-seq OCR with Spearman correlation superior to 0.75 (Supplementary Fig. 2A).”

Supplementary Fig. 2:

A. Cross-validated Cellformer performance stratified by cell type.

B. Cellformer outperforms the baseline models across cell types.

(2) Since lack of ATAC-Seq data of cell type proportion-known mixture samples, it is reasonable and essential to conduct evaluations on comprehensive synthetic datasets. However, the authors only generate via the setting ‘5000 pairs of synthetic bulk and cell-type specific bulk were generated from each subject, composed of a random number of

cells ranging from 100 to 800'. What happens if certain cell types are missing from the bulk or single-cell data?

Response: We conducted additional evaluations to determine the effect of missing or new cell types and added our findings (new **Supplementary Fig. 3**) to the Results section. New text is: "In real-life scenarios, the cell type composition of bulk tissue remains unknown. For instance, a rare cell type can be missing or a new (unidentified) cell type can emerge in bulk tissue. In both scenarios, Cellformer is minimally affected by the presence or absence of one cell type, as there are no significant differences in the model's performance across different cell types (**Supplementary Fig. 3**)."

A Scenario: a cell type is missing from bulk ATAC-seq

B Scenario: a new cell type appears in bulk ATAC-seq

Supplementary Figure 3: Cellformer evaluation in real-life scenarios.

A. Simulation of a real-life scenario where a cell type is absent in bulk tissue. Boxplots illustrate Cellformer performance when two rare cell types, OPCs and MIC, are

missing from the pseudo bulk samples used to test the model. Top panel represents the absence of OPCs, bottom panel represents the absence of MIC.

- B. Simulation of a real-life scenario where a new cell type, previously unseen by the model, emerges in bulk tissue. Boxplots illustrate Cellformer performance when two rare cell types, OPCs and MIC, are intentionally removed from the synthetic pseudo bulk samples during training and added at testing. Top panel represents the absence of OPCs; bottom panel represents the absence of MIC.

(3) How does Cellformer perform on cells that have only 5% and 2% out of all the samples? One of the main unresolved issues is a good performance on rare cell types.

Response: We performed additional analysis to determine Cellformer performance on rare cell types (OPCs and microglia) (panel A of new **Supplementary Fig. 4**) and added text in the Results section: "We evaluated Cellformer's performances on pseudo bulk samples made with different percentages of cell type-specific cells. We observed a slight decline in Cellformer's performance when cells make up less than 10% of the total bulk cells. For biologically rare cells such as OPCs (constituting less than 3% in white matter) or microglia (constituting less than 10% in brain), Cellformer achieves an average Spearman correlation of 0.7 when deconvoluting pseudo bulk data, with OPCs which account for less than 3% of the overall composition. Similarly, an average correlation of 0.68 is achieved when deconvoluting pseudo bulk samples containing less than 10% of microglia (**Supplementary Fig. 4A**)."

Supplementary Figure 4A: Cellformer performance evaluated using synthetic pseudo bulk data, with varying percentages of cells per cell type.

(4) Since Cellformer uses reconstruction loss to aid in learning, how does the sample size of the data affect the performance?

Response: Thank you for this important question. We performed additional experiments (new panel D of **Supplementary Fig. 2**) to determine the effect of sample size on performance and added the following text in the Results section: "As a learning-based algorithm, Cellformer relies on snATAC-seq to learn cell type profiles. Using our proposed synthetic pseudo bulk data generation strategy, we show that Cellformer can be trained effectively with a limited number of snATAC-seq samples, with minimal effect of sample size on its performance (Krustal-Wallis P-value 0.98)(**Supplementary Fig. 2D**)."

Supplementary Figure 2D: Cellformer performance when trained using varying numbers of snATAC-seq samples. To limit confounders, we restricted our analysis to samples from the same brain region (SMTG). Cellformer is weakly impacted by sample size (Kruskal-Wallis P-value 0.98).

(5) The considered benchmark methods are limited and are rather simple. Moreover, the authors only compare the performance of Cellformer to other benchmarks in Figure 2, but failed to incorporate any comparison in Figure 3 and Figure 4. From my knowledge, the idea of cell-type specific ‘deconvolution’ is not new in this area; maybe the author should consider conducting comprehensive evaluations with existing works like scDeconv, DeconPeaker, CoRE-ATAC, and references therein.

Response: We added new text to clarify that in contrast to previous methods to deconvolute bulk sequencing data, Cellformer is not designed to predict cellular abundance in bulk tissue, but rather to “fully” deconvolute bulk ATAC-seq data, *i.e.*, retrieve the chromatin accessibility for each cell type.

We also compared Cellformer to two established deconvolution methods (new panel C in **Supplementary Fig. 2**). We added the following text to the Results: “Current state-of-the-art deconvolution methods such as scDeconv (Liu 2022), DeconPeaker (Li et al. 2020), BayesPrism (Chu et al. 2022) and CIBERSORT (Newman et al. 2015), rely on a cell type-specific expression matrix, using the most highly distinguished markers per cell type, to predict the cellular composition of bulk tissue. In contrast, Cellformer predicted cell type-specific expression of more than 41954 OCR, which is 2.5 fold more output than established deconvolution methods (**Supplementary Fig. 2C**). This enables more comprehensive downstream analysis of biological systems at the cell type level and highlights the ability of more extensive deconvolution to gain deeper insight from bulk data.”

Supplementary Figure 2C: Barplot comparing the number of generated features among CIBERSORT (Newman et al. 2015), BayesPrism (Chu et al. 2022) and Cellformer.

Reviewer #2 (Remarks to the Author):

Comments for Author

General comments:

In this manuscript, the authors have developed a novel approach for deconvoluting bulk ATAC-seq data to infer cell type-specific ATAC-seq expression as well as sample composition. The authors use a reference-based deep learning method, which they show outperforms other machine learning methods for this task. On the biological side, they compare the cell type-specific expression between resilient to AD (RAD), normal control, and AD dementia (ADD) individuals, and find that open chromatin regions (OCRs) differences between RAD and the other groups are strongly localized to the hippocampus region, and identify key pathways and cell types whose genes may have altered accessibility.

Overall, the manuscript presents a promising approach for deconvoluting bulk ATAC-seq data to infer cell type-specific ATAC-seq expressions. However, the manuscript still leaves some key questions unanswered, and some additional validation is required to fully support the authors' conclusions.

Major comments:

1. The schematic of the model in Figure 2 could use a little more detail. In particular, it shows the reconstruction of cell type-specific profiles, but does not show where/how the cell type composition measures are derived. Some of these details are in the methods, but it would be good to show them up front, particularly for non-experts.

Response: We expanded **Figure 2a** to show how the cell type-specific profiles used for reconstruction were derived, with additional explanation in the legend.

Figure 2a A synthetic dataset of simulated bulk samples was generated from previously published single-cell ATAC-seq from 13 normal controls(Corces et al. 2020). Cell type-specific pseudo bulk samples were generated by aggregating snATAC-seq data, revealing the ground-truth cell type-specific composition. The simulated cell-specific pseudo bulk samples were further aggregated to generate pseudo-bulk samples, which are Cellformer's input.

2. It would be helpful if the authors could provide more details about the batch effects that were adjusted for in their processing pipeline. This could include information about the experimental design, such as the sequencing batches, sample collection dates, and any other relevant factors that may introduce batch effects. The methods used to adjust for batch effects should also be described in more detail. For example, did the pre-processing or model incorporate any batch normalization techniques, such as ComBat or SVA, to adjust for batch effects? If so, they should describe how these methods were applied and any parameters that were used in the process. In addition, the authors should include quantification of batch mixing (e.g. LISI scores or other measures on the plots shown in Fig. S5) to show that batch effects were essentially removed by the model.

Response: We added information about data processing and batch mixing quantification to the Methods section: "Harmony batch normalization was applied on snATAC-seq to

ensure robust peak calling as previously described (Corces et al, 2020, Granja et al. 2021). No batch normalization was further applied on the input of the model, since additional analysis suggests that Cellformer removes batch effects while preserving biological variations, *i.e.*, kBet = 0.79 +/- 0.14, NMI= 0, ARI = -0.02 +/- 0.001, cLISI = 1, iLISI = 0.58 +/- 0.02 across the brain regions (Luecken et al. 2021).”

3. While the Cellformer approach appears to be effective at deconvoluting bulk ATAC-seq data to infer cell type-specific ATAC-seq expressions, it may not be able to fully capture the cell type subpopulations that are available through single-cell sequencing methods. Is Cellformer limited to the resolution of major cell classes in the brain, or can it also recapitulate OCRs in distinct subtypes within a class e.g. SST+ versus PVALB+ neurons in cortex and hippocampus? Although it is not necessary to test every subtype, it is possible that the Cellformer approach could be rerun with finer resolution clusters in the training set to identify composition/OCR profiles of subpopulations of cells within bulk sequencing data?

Response: Based on this excellent comment, we reran Cellformer at a finer resolution (panel B in new **Supplementary Fig. 4**) and included the following text in the Results section: “Finally, although we primarily focused on the major brain cell classes in this study, we also assessed the performance of Cellformer in accurately capturing OCRs in specific subclasses such as SST+ and PVAL+ inhibitory neurons (**Supplementary Fig. 4B**)”.

The following limitation is added in the Discussion: “In this study, we focused on the 6 major brain cell classes. Although we demonstrate Cellformer’s ability to deconvolute at a lower resolution (**Supplementary Fig. 4B**), we noticed that increasing the number of output cell types results in a significant rise in computational complexity. This limitation restricts the number of deconvoluted cell types that can be effectively handled. To overcome this challenge, we plan to implement and validate strategies such as hierarchical training or cell type prioritization functions, which will expand the capabilities of Cellformer.”

Supplementary Figure 4B: Cellformer performances when trained to deconvolute bulk ATAC-seq data at a lower resolution.

4. The authors used ChIPseeker to annotate peaks to genes may not be the most precise method, as it relies on assigning regions to genes based on proximity. This can lead to false positives or false negatives in the annotation of peaks to genes, as well as potential ambiguities in the assignment of peaks to specific genes. How robust are the biological results regarding RAD robust to different peak-to-gene annotations?

Response: Thank you for this important point. We expanded the Methods section as follows to justify our choice of ChIPseeker and address these concerns: “[... *genomic OCR annotation*] using default parameters following ATAC-seq data processing guidelines and Harvard bioinformatics recommendations (Yan et al. 2020; Gaspar 2019). One of the main issues in epigenetic analysis is the lack of consensus between annotating tools (Kondili et al. 2017). We therefore compared ChIPseeker to the annotations given by ArchR, developed for ATAC-seq data analysis. Overall, ChIPseeker and ArchR agreed on 60% of the peaks, including complete (100%) agreement between peak-to-gene annotations of RAD-specific promoters, from which our biological insights were drawn. In particular, similar gene ontology enrichment is observed using both tools.”

5. It is important for the authors to explore the biological implications of the observed OCR differences between RAD and the other groups. While the authors have performed functional annotation and pathway analysis to identify enriched biological pathways or functions that are associated with the differentially expression open chromatin regions. The authors could perform further validation studies to confirm the significance of the observed OCR differences, for example through staining or ISH of a few predicted cell type-specific genes in the hippocampus tissue. Overall, validating the key biological implications of the observed OCR differences would provide a more complete picture of the mechanisms underlying resilience to Alzheimer's disease, and could have important implications for the development of novel therapeutic approaches.

Response: : To address the biological implications and underlying mechanisms, we compared our findings to quantitative proteomics data generated from the same samples for another study (Merrihew et al. 2023). We believe this approach offers a more robust validation than more qualitative methods like immunohistochemistry. The following validation results (new panels D-F in Supplemental Figure 8) were added in the Results section: "We conducted additional validation of RAD epigenetic signatures by using proteomics data previously collected on the same samples (Merrihew et al. 2023). A weak agreement is observed between proteomic expression and ATAC-seq accessibility with a Pearson correlation of -0.001 (Supplementary Fig. 8D). Only 8% of OCR-related genes show overlap with expressed proteins (Supplementary Fig. 8E). Similar results are observed with RAD-specific OCR related genes, with 4 out of 40 (10%) genes overlapping with expressed proteins. However, in contrast to the overall sample, our analysis reveals that two (50%) protein coding genes (VDAC2 and PGBD5) exhibited significant upregulation in RAD at both epigenetic and proteomic levels (Supplementary Fig. 8F)."

Supplementary Figure 8D-F:

D. Correlation between the mean protein expression and ATAC-seq accessibility from NC in HIPP. We used promoters associated with protein coding genes.

E. The overlap between OCR-related genes and expressed proteins in HIPP.

F. Proteomic expression levels of PGBD5 and VDAC2 which are also upregulated in RAD at the epigenetic level.

6. In Fig. 2G, the cross-comparison between model predicted OCRs and SEA-AD OCRs shows some off-diagonal correlations. These are likely biological, given cell type similarity, but it would be good to have "baseline" heatmaps of the same type showing cross-cell type correlations within the SEA-AD data alone and the Cellformer predictions alone. In the likely event that this looks very similar to the existing Fig. 2G, it would strength the argument that Cellformer can deconvolute cell types with a range of similar OCRs.

Response: As suggested, we constructed baseline heatmaps (new panel B in Supplementary Figure 7) to complement our validation and added the following text in the Results section: "These inter-cell type correlations were also observed within snATAC-seq and deconvoluted ATAC-seq mean profiles, suggesting that Cellformer can deconvolute cell types with a range of similar OCRs (Supplementary Fig. 7B)." The following figure was added in the Supplementary Materials:

Supplementary Figure 7:

B. snATAC-seq and deconvoluted ATAC-seq mean profile autocorrelation matrices.

Minor comments

1. Figure 2 would be improved by increasing the asterisk size in panels C, E and G.

Response: Corrected

2. It would be highly informative if the authors could provide the age, sex and ethnicity information of the RAD, NC and ADD donors the authors used in this study. Although Figure S5C shows there is no strong sex effect at the top cell type level, it would be good to know the balance between sexes, as well as the age distributions between the three groups.

Response: We added this information as new Supplementary Figure S1 and cite it in the Introduction and Methods: "...sex and age range matched..(Supplementary Fig. 1)"

	Female	Male	Age (+/- sd)	White	Black/African American	Asian
NC	2	2	80.33 +/- 12.96	3	1	0
RAD	6	6	82.25 +/- 7.63	10	1	1
ADD	8	11	89.95 +/- 6.08	17	0	2

Supplementary Figure 1: Dataset overview: sex, age and ethnicity and sex across groups. ADD = Alzheimer's disease dementia, RAD = resilient to Alzheimer's disease, NC = normal control, ns = not significant.

3. In Figure S5C, suggestion to replace "Expired Age" with "Age at death"

Response: Corrected (now Figure S8C)

4. In the methods (line 461): “For each pair in the synthetic dataset, a random number of same type single nucleus within was merged using single nucleus data from one snATAC-seq, preserving regional and individual diversity in our synthetic dataset”, it seems there are some words missing. “single nucleus samples”? “one snATAC-seq donor”?

Response: We rephrased this sentence to make it clearer: “Each pair in the synthetic dataset was created by first sampling and aggregating a random number of the same type of single nuclei from a sample’s snATAC-seq results in order to create synthetic cell type-specific pseudo bulk samples that preserve regional and individual diversity in our synthetic dataset. Then, corresponding synthetic bulk ATAC-seq data were created by aggregating the generated cell type-specific pseudo bulk samples from the 6 cell types.”

REVIEWERS' COMMENTS

Reviewer #1 (Remarks to the Author):

Most of my previous comments have been resolved. I do not have further comments.

Reviewer #2 (Remarks to the Author):

The authors have resolved my questions, I have no additional questions.